# Estrogen Modulates Epithelial Breast Cancer Cell Mechanics and Cell-to-Cell Contacts

**DOI:** 10.3390/ma14112897

**Published:** 2021-05-28

**Authors:** Barbara Zbiral, Andreas Weber, Jagoba Iturri, Maria d. M. Vivanco, José L. Toca-Herrera

**Affiliations:** 1Institute for Biophysics, Department of Nanobiotechnology, University of Natural Resources and Life Sciences, Muthgasse 11, 1190 Vienna, Austria; barbara.zbiral@boku.ac.at (B.Z.); Jagoba.iturri@boku.ac.at (J.I.); 2CIC bioGUNE, Basque Research and Technology Alliance, BRTA, Bizkaia Technology Park, 48160 Derio, Spain; mdmvivanco@cicbiogune.es

**Keywords:** cell mechanics, breast cancer, estrogen, AFM, viscoelasticity, E-cadherin, confocal microscopy

## Abstract

Excessive estrogen exposure is connected with increased risk of breast cancer and has been shown to promote epithelial-mesenchymal-transition. Malignant cancer cells accumulate changes in cell mechanical and biochemical properties, often leading to cell softening. In this work we have employed atomic force microscopy to probe the influence of estrogen on the viscoelastic properties of MCF-7 breast cancer cells cultured either in normal or hormone free-medium. Estrogen led to a significant softening of the cells in all studied cases, while growing cells in hormone free medium led to an increase in the studied elastic and viscoelastic moduli. In addition, fluorescence microscopy shows that E-cadherin distribution is changed in cells when culturing them under estrogenic conditions. Furthermore, cell-cell contacts seemed to be weakened. These results were supported by AFM imaging showing changes in surfaces roughness, cell-cell contacts and cell height as result of estrogen treatment. This study therefore provides further evidence for the role of estrogen signaling in breast cancer.

## 1. Introduction

Breast cancer is a major cause of cancer-related death for women and, despite all advances, there are still many unresolved research questions and unmet clinical needs [1]. Estrogens are a group of steroid hormones that play a considerable role in regulating the growth of normal breast epithelium, while excessive exposure to estrogen is associated with increased risk of breast cancer [2]. While estrogen receptor (ER)-positive breast tumors are generally considered less aggressive and have better prognosis, they can nevertheless develop drug resistance and undergo metastasis [3]. Epithelial-mesenchymal transition (EMT) programs play essential roles in normal mammary gland development, and are also associated with the enhanced migratory and invasive capacities characteristic of breast cancer lesions when they become highly aggressive and invasive tumors [4]. Estrogen has been shown to promote reversible EMT [5] and regulate the activity of the extracellular proteolytic molecular network [6]. Furthermore, estrogen signaling contributes to deregulation of adherens junctions (AJ), which leads to altered cell-to-cell contacts [7]. However, the full involvement of estrogen on breast tumor development and progression is yet to be completely understood and remains somewhat controversial.

During cancer progression, malignant breast cancer cells accumulate properties distinctive from those of healthy cells, not only in their morphology and biochemistry, but also in their mechanical cell properties. The latter are of particular significance with regards to cell migration and invasiveness, which are considered hallmarks of cancer progression. Within a cell, cytoskeleton dynamics account for many measurable cell mechanical properties. Thus, cytoskeletal rearrangements—which may arise from cancer-associated changes in cell biochemistry—lead to measurably altered mechanical properties in cancer cells. Additionally, membrane profile changes, altered adhesive properties, and gradual loss of cell-to-cell contacts contribute to variations in cell mechanics during breast cancer progression [8,9,10].

Over the past decades, alongside emerging insights into the biological significance of cell mechanics, the field of mechanobiology has gained steady traction [11,12,13]. This term refers to the sensing and processing of mechanical information from the cell environment and how this influences certain signaling pathways (mechanotransduction) [14,15,16,17,18]. Nowadays, a diverse array of methods for studying the mechanical properties of biological samples such as single cells, multicellular aggregates or tissue sections is available [19,20,21]. Among these, atomic force microscopy (AFM) based force-spectroscopy has enjoyed popularity for characterizing cell mechanics [11,22,23,24,25,26,27]. Here, a cantilever probe of known properties controlled via a piezoelectric element indents a cell while a photodiode simultaneously records the deflection of a laser from the reflective backside of the probe. AFM force-spectroscopy retains multiple advantages over other established methods for studying cell mechanics [11,28,29]. A single AFM force-spectroscopy experiment yields multiple distinct mechanical and viscoelastic parameters, such as the apparent elasticity relaxation times, compressive moduli and viscosities [30,31,32,33,34,35].

In this study, we performed stress relaxation experiments on ER-positive MCF-7 breast cancer cells treated with 17-β-estradiol (estrogen, E_2_). Mechanical measurements were performed using nm-sized tips, as well as µm-size particles, employing stress relaxation measurement conditions. In addition, E-cadherin localization was observed by fluorescence microscopy.

Overall, we found that addition of estrogen significantly softens the cancer cells compared to untreated controls, suggesting that estrogen alters cell-to-cell contacts in ER-positive breast cancer cells and induces measurable changes in cell mechanics.

## 2. Materials and Methods

### 2.1. Cell Culture and AFM Sample Preparation

MCF-7 breast cancer cells were obtained from the American Type Culture Collection (ATCC) and were grown in high glucose DMEM (4.5 g/L glucose) with stable glutamine, supplemented with 10% fetal bovine serum (FBS) and 1% penicillin/streptomycin. Cells were cultured routinely and passaged when reaching a confluence of 80%. For experiments, cells with passage numbers ranging from 5 to 30 were used. For AFM sample preparation, glass slides were rinsed with ethanol and plasma-cleaned for 1 min. Then, 5 × 10^4^ cells were seeded on the glass slides and grown for 24 h prior to treatments. In addition, phenol red-free DMEM supplemented with dextran-coated charcoal (DCC, Sigma) treated FBS (to remove steroid hormones from the serum) was used to test the effect of medium without hormones.

For experiments, cells were either grown in the above specified DMEM (called CTL DMEM) or DCC-treated DMEM without phenol red (called CTL HF). Estrogen stocks were prepared by dissolving estrogen (Sigma, St. Louis, MO, USA) in DMSO at a concentration of 1 mg/mL. Cells were treated for 24 h using an estrogen concentration of 100 nM (referred to as DMEM E_2_ and HF E_2_).

### 2.2. Cell Fixation, Permeabilization and Immunostaining

Cells were grown for 24 h at a density of 5 × 10^4^ cells/mL in Ibidi µ-slides either in DMEM or HF medium and then treated for 24 h with 100 nM E_2_. Cells were fixed with 4% paraformaldehyde for 10 min at RT and then permeabilized using 0.1% Triton X-100 (diluted in PBS) at RT for 15 min. This was followed by a blocking step using 2% BSA in PBS solution at RT for 1 h. For immunostaining, cells were incubated with primary anti E-cadherin mouse monoclonal antibody (13-1700), diluted 1:1000 in 1% BSA (2 µg/mL final concentration) at 4 °C over-night. Then, a secondary goat anti-mouse antibody (A32723), conjugated with Alexa Fluor 488 was added in a 1:1000 dilution in 1% BSA (2 µg/mL) together with phalloidin labeled with Alexa Fluor 555 (diluted 1:40 in 1% BSA) and incubated for 45 min at RT. Subsequently, nuclei were stained using Hoechst 33342 (diluted 1:1000 in PBS). In between all steps, samples were washed multiple times using PBS. Samples were kept at 4 °C, protected from light and measured as soon as possible. At least two independent samples with the respective controls were prepared. All materials were purchased from Thermo Fisher Scientific (Waltham, MA, USA).

### 2.3. Confocal Laser Scanning Fluorescence Microscopy

For CLSM measurements, cultured and fixed cells were washed and kept in PBS for the duration of the measurements. A Leica DMi8 microscope stand equipped with a SP8 scanning head, a 405 nm laser and a tunable white light laser (470 to 670 nm) equipped with 2 HyDs and 2 PMTs as detectors was used. A 40× oil immersion objective was employed. Excitation and emission wavelengths were optimized for the different staining dyes. Images were post-processed using the Fiji distribution of ImageJ [36].

### 2.4. Atomic Force Microscopy—Imaging

For AFM imaging, cells were seeded onto plasma-treated glass slides and fixed as described above. A JPK Nanowizard III system (Bruker, Germany) was used to perform imaging in contact mode. Triangular MLCT cantilevers (Bruker, Germany) with nominal spring constants of 0.01 and 0.02 N/m, a resonance frequency of 7 kHz in air and a tip radius of 20 nm were used. Spring constants of the cantilevers were calibrated using the thermal noise method [37]. Imaging was performed with force set points between 0.4 and 0.8 nN, line rates of 0.5–0.8 Hz, scan sizes ranging from 75 µm × 75 µm to 100 µm × 100 µm and a resolution of 512 × 512 pixels. Those measurements were performed in PBS at room temperature. Images were analyzed using JPKSPM software (Version 6.1.159). First, a polynomial surface was subtracted from the image and then a low-pass filter was applied. The height of at least 25 cells was determined from the AFM images.

### 2.5. Surface Roughness Determination

From the AFM images, both the arithmetic average roughness *R_a_* and the root mean square roughness *R_q_* (also known as *R_rms_*) were determined. They are defined as
(1)$$R_{a}=\frac{1}{n_{x}n_{y}}\overset{n_{x}}{\underset{i=1}{\sum }}\overset{n_{y}}{\underset{i=1}{\sum }}\left(Z\left(i,j\right)-\bar{Z}\right)$$
(2)$$R_{q}=\sqrt{\frac{1}{n_{x}n_{y}}\overset{n_{x}}{\underset{i=1}{\sum }}\overset{n_{y}}{\underset{i=1}{\sum }}{\left(Z\left(i,j\right)-\bar{Z}\right)}^{2}}$$
with *n_x_* and *n_y_* being the number of pixels on the *x*- and *y*-axis, *i* and *j* correspond with to the index of the pixel in the x and y direction and *Z*(*i*, *j*) being the corrected height value of the sample at a respective pixel. Both values were determined by taking five randomly selected regions of an area of 5 µm^2^ (32 × 32 pixels) in the central region of each cell.

### 2.6. Atomic Force Microscopy—Force Spectroscopy

AFM force spectroscopy measurements were performed using a JPK Nanowizard III head with a CellHesion module mounted on top of an inverted optical microscope (Axio Observer Z1, Zeiss, Oberkochen, Germany) in a temperature-controlled liquid sample stage at 37 °C in Leibovitz L15 medium. Silica particles with a diameter of 10 µm were glued to tipless, triangular cantilevers as described previously [38]. Either cantilevers with a four-sided pyramidal tip (nominal radius of 10 nm, spring constant of 0.12 N m^−1^ and resonance frequency of 23 kHz in air) or with a 10 µm silica particle were used (DNP-S and NP-O, cantilever B). Cantilevers were cleaned with EtOH, dried with N_2_ and cleaned with 30 min of UV/O. Prior to measurements, the spring constants of cantilevers were calibrated by acquiring force-distance-curves on a stiff substrate (glass) and using the thermal noise method by applying the equipartition theorem. After mounting the AFM head to the sample stage, the system was left to equilibrate for 30 min. Figure 1 shows a schematic drawing of the AFM measurement set-up.

The cells were approached at a loading rate of 5 µm/s with a maximum force set point of 2 nN (settings were determined as described recently [38,39]). After this, a stress relaxation period of 10 s was applied. The cantilever was retracted with a loading rate of 5 µm/s. The z-range of approach and retract was 50 µm and the data was recorded with a sample rate of 1024 Hz. Each cell was indented 5 times at the central, nuclear region. Per sample, at least 25 cells were measured and overall at least 3 samples per condition were measured. A sample was kept in the sample stage for a maximum of 4 h. No changes in cell appearance were observed during this time span.

### 2.7. Data Analysis of Force Spectroscopy Curves

All analysis of curves was performed using the R afmToolkit, recently developed by Benitez et al. in our laboratory [40]. Briefly, force-distance curves were extracted and imported into the toolkit, then contact and detachment points were determined using the algorithm already described before with optimized parameters [41]. Then, the baseline was corrected, a zero-force point was determined and the deformation of the sample was calculated according to
(3)$$\delta_{s}=Z_{p}-\delta_{c}=Z_{p}-\frac{F}{k_{c}},$$
modeling the cantilever as an ideal elastic spring with *Z_p_* being the z-position of the piezo, $\delta_{s}$ and $\delta_{c}$ the deformation of the sample and the cantilever respectively, *F* the measured force and *k_c_* the spring constant of the cantilever. The cantilever deflection is measured by the position of a laser spot on a four-quadrant photodiode. From both the cantilever stiffness and the measured stiffness, the stiffness of the sample can be determined according to
(4)$$k_{s}=\frac{k_{m}k_{c}}{k_{m}+k_{c}}.$$

#### 2.7.1. Elastic Properties

In a first approximation to the mechanical properties of the samples, the Sneddon extension of the Hertz model for linear ideal elasticity was used to determine the apparent Young’s modulus *E_app_* of the cells. This was done for either a four-sided pyramidal indenter geometry (Equation (5)) or a paraboloid contact profile (Equation (6)) as
(5)$$F=\frac{E_{app}}{1-\nu^{2}}\frac{\tan\alpha }{\sqrt{2}}\delta^{2},$$
(6)F=43RcEapp1−ν2δ32,
with ν as the Poisson’s ratio of the cells (set to 0.5, modeling the cells as incompressible), *α* the half-opening angle of the pyramid, *δ* the indentation (also known as $\delta_{s}$) and $R_{c}$ the radius of the spherical particle (5 µm). An indentation of 500 nm of the approach curve was used to determine the Young’s modulus, since the assumptions made for using this model (linear elasticity, isotropy, infinite half-space) are met for shallow indentations below 10% of sample height [42].

#### 2.7.2. Viscoelastic Properties

Due to their complex structure, cells behave as viscoelastic bodies [43,44,45]. Stress relaxation experiments were used to determine the response of cells to a given constant strain over time (strain ramping assumed instant). Curve fittings were performed by the R afmToolkit employing a non-linear Levenberg–Marquardt algorithm with optimized start parameters.

The ratio of stress σ to strain ε can be expressed in the Laplace domain as the rigidity/shear modulus $\tilde{G}$
(7)$$\frac{\tilde{\sigma }}{\tilde{\varepsilon }}=2\tilde{G}\left(s\right),$$
which is connected to the Young’s modulus in the Laplace domain [46] through
(8)$$\tilde{G}\left(s\right)=\frac{\tilde{E}\left(s\right)}{2\left(1+\nu \right)}.$$

Using stress relaxation conditions, the expression in Equation (8) can be combined with Equations (5) and (6) to determine viscoelastic properties, as proposed by Darling et al. [33] for a combination of Hertzian mechanics with a standard linear solid model. Equations (5) and (6) can be approximated by using a Heaviside step function *H*(*t*)
(9)$$F\left(t\right)=C\frac{E\left(t\right)}{1-\nu^{2}}H\left(t\right)$$
with the constant *C* depending on the geometry of the indenter, defined as
(10)Cspherical=43Rcδ032
for spherical indenters and
(11)$$C_{pyramidal}=\frac{1}{\sqrt{2}}\tan\alpha {\delta_{0}}^{2}$$
for a four-sided pyramid. Note that the term $\delta_{0}$ refers to the indentation value that is kept constant during the stress relaxation phase. Transforming Equation (9) to the Laplace domain yields
(12)$$\tilde{F}\left(s\right)=C\frac{1}{1-v^{2}}\frac{\tilde{E}\left(s\right)}{s}.$$

Substitution of $\tilde{E}\left(s\right)$ with Equation (8) then leads to
(13)$$\tilde{F}\left(s\right)=C\frac{2}{1-v}\frac{\tilde{G}\left(s\right)}{s}.$$

Transforming Equation (13) back to the time domain yields a solution to the viscoelastic model. The analytical solution for $\tilde{G}\left(s\right)$ depends on the structure of the employed viscoelastic model. In this work, we employ a generalized Maxwell model using two Maxwell branches. The relaxation modulus $\tilde{G}\left(s\right)$ in the Laplace domain can be described by an empirical Prony series as
(14)$$\tilde{G}\left(s\right)=G_{\infty }+\overset{N}{\underset{i=1}{\sum }}G_{i}\frac{\tau_{i}s}{1+\tau_{i}s},$$
where $G_{\infty }$ is the equilibrium shear modulus, $G_{i}$ are the moduli of springs in the respective Maxwell element and $\tau_{i}$ the relaxation times following
(15)$$\tau_{i}=\frac{\eta_{i}}{G_{i}}.$$

Transforming Equation (14) back to the time domain yields
(16)$$G\left(t\right)=G_{\infty }+\sum G_{i}\exp\left(-\frac{t}{\tau_{i}}\right),$$
and the instantaneous shear modulus is then given by
(17)$$G_{inst}=G_{\infty }+\sum G_{i}.$$

Combing Equation (16) for a two element Maxwell model with Equations (8) and (13) transformed to the time domain leads to
(18)$$F\left(t\right)=\frac{C}{1-\nu }\left(E_{\infty }+\overset{2}{\underset{i=1}{\sum }}E_{i}\exp\left(-\frac{t}{\tau_{i}}\right)\right),$$
with the constants defined as in Equations (10) and (11).

### 2.8. Statistical Analysis

All statistical analysis was performed using Origin Pro 2018 (by OriginLab, Northampton, MA, USA). Outliers were determined via a Grubbs test and removed. For determining statistical significance of differences between groups, normality of samples was checked with a Kolmogorov–Smirnoff test for normality, followed by a one-way ANOVA (significance level always set as 0.05). Non-normally distributed samples were compared using a Kruskal–Wallis ANOVA. Significances are reported as * for *p* < 0.05, ** < 0.01 and *** < 0.001.

## 3. Results

### 3.1. Estrogen Leads to Softening of Breast Cancer Cells

#### 3.1.1. Elastic Properties from Indentation Curves

To determine the effect of estrogen on cell mechanics, the apparent elastic modulus was determined using Hertzian mechanics both for measurements performed with tips or with 10 µm diameter particles. Treatment of MCF-7 cells with 100 nM E_2_ led to significant softening (Figure 2). As expected, considering the contact profile and the literature, the Young’s modulus is in the range of a few kPa when using a tip and in the range of a few hundred Pa when using a particle [20,38,39,47].

Interestingly, when tips were used, measurements were similar in the control cells, independent of the culture medium used. In contrast, measurements with particles yielded higher elasticities in HF CTL than in DMEM CTL medium, suggesting that in hormone-depleted conditions cells were stiffer. Regarding measurements with tip indenter (10 nm nominal tip radius), estrogen treatment leads to a reduction of cell stiffness of around 17% in DMEM and 30% when growing cells in hormone-free medium. For particle measurements, in DMEM a reduction of Young’s modulus by 11% can be seen while for the HF media, a reduction of 28% is observable. Growing cells in hormone-free medium leads to an increase of the elastic modulus and therefore, to a more pronounced reduction when treated with estrogen.

In addition, the indentation *δ* at the maximum load of 2 nN was evaluated. For tip measurements, indentations of around 2 to 3 µm were recorded, while for particles, values of approximately 1.25 µm were measured. This is due to the particle possessing a larger surface area than the tip. Therefore, for the particle to reach similar indentations, a higher force would be needed than for tips. For the tip measurements, an increase in the maximum indentation when adding estrogen is shown in Figure 1. Again, culturing cells in HF medium leads to a more pronounced effect of estrogen addition. In contrast, the indentations with a particle showed no significant differences between either control or treated groups. This probably is due to the particle measuring the averaged mechanical properties of the cell (and all the indented cellular substructures), while the tip is susceptible to highly localized differences. In other words, the pressure is more distributed when using a particle.

#### 3.1.2. Viscoelastic Properties from Stress Relaxation Experiments

Cells are known to behave as complex hierarchical viscoelastic materials due to the different components of the cell (such as actin cortex, membrane, cytoplasm, nucleus and other cytoskeletal networks). To test the influence of estrogen on cell viscoelastic properties, stress relaxation curves at a load of 2 nN were recorded. Note that this corresponds to an indentation of approximately 2 µm when using a tip and around 1.25 µm with a particle as indenter. A double-exponential decay was best suited to fit the full relaxation behavior (see Appendix A). A generalized Maxwell model was used to calculate cell mechanical properties, such as the equilibrium modulus *E_∞_*, the moduli of the springs in the Maxwell elements *E*_1_ and *E*_2_, the instantaneous modulus *E_inst_* and the viscosities *η* of the Maxwell elements.

Figure 3 shows the influence of estrogen on the viscoelastic properties of MCF-7 cells measured either with tips (a) or with particles (b). Table 1 shows the derived statistical values (full statistical analysis can be found in the Appendix A). As before, moduli estimated from the tip measurements are higher than those for particle measurements. Cells cultured under hormone-free conditions showed increased moduli values (Figure 3a,b). Treatment with estrogen led to a significant reduction of viscoelasticity in both media, which was strongest in cells cultured with HF medium.

Analysis of the viscoelastic properties with a particle showed a reduction in viscoelastic moduli when cells were treated with estrogen (Figure 3b), indicating that estrogen leads to a softening of the cells. The equilibrium modulus was the most affected value in cells grown in DMEM in the presence of estrogen (minus 28%, compared to a reduction of 18% roughly for the other moduli). For both media, tip and particle measurements led to comparable results. Cell culture in hormone-free medium yielded more pliable cells in the presence of estrogen, but here the differences are less pronounced compared to tip measurements (reduction of 17, 8, 8 and 13.5% for the respective moduli).

Finally, viscosities were derived from the relaxation times obtained from the fittings and the viscoelastic moduli. Two distinctive relaxation times, one in the range of a few hundred milliseconds and one of a few seconds were determined, as described in prior work [31]. In literature, the presence of two relaxation times is thought to originate from different structures inside the cell, and the larger is normally connected to the cytoskeleton. Table 1 as well as Figure 4 show the determined viscosity values. Values were approximately five times higher when calculated for tips than for particles (Table 1). This difference is due to multiple reasons; on the one hand, the tip has a smaller contact area than the particle, thus less cell volume is deformed. On the other hand, respective indentations at the same force differ (2 µm for the tip and 1.25 µm for the particle), presumably due to probing different cell compartments. Viscosity in cells cultured in DMEM, when measured with tip, was hardly affected by estrogen addition (Figure 4a). In contrast, the viscosity of cells grown in hormone-free medium was strongly reduced by estrogen addition. Both viscosities are significantly smaller for measurements performed with particles after estrogen supplementation, irrespective of medium (Figure 4b). The viscosities are in range with those reported for cells in literature [20,31,33,48].

### 3.2. Surface Roughness and Cell Height Increases Due to Estrogen Treatment

To evaluate changes in the three-dimensional morphology of MCF-7 cells resulting from estrogen stimulation, we performed contact mode AFM imaging. Representative images can be seen in Figure 5, with the left column showing the false color height image and the right column the respective error image. We imaged at least five different cell clusters.

Cell height appeared to increase slightly when treated with estrogen (Figure 5b,d). In addition, the surface roughness of the cells around the nuclear region appeared to be increased. To quantify these effects, surface roughness and cell height were determined from at least 25 AFM height images of cells. Results of this analysis are summarized in Table 2. Addition of estrogen led to a significant rise in both cell height and surface roughness of cells cultured in DMEM medium. Both cell height and roughness were higher in cells cultured in HF than in DMEM medium. Addition of estrogen increased these parameters, but not to the same level as those for DMEM and estrogen treated cells, suggesting that perhaps something else in the serum was contributing to these effects.

### 3.3. Estrogen Changes E-Cadherin Distribution

To further analyze the influence of estrogen on MCF-7 cells, fluorescence microscopy was performed by staining the nuclei, E-cadherin and actin. Representative images of these experiments can be seen in Figure 6. Control cells showed a well-defined, cobblestone-like architecture, with E-cadherin strongly localized on the cell rims, enhancing cell-cell contacts. In addition, well-defined actin structures are visible along the rim of the cells. Addition of estrogen led to a wider distribution of E-cadherin throughout the cytoplasm. Furthermore, cell borders appeared disrupted, including some small micro-clustering of E-cadherin and reduced cell-cell contact, suggesting that estrogen is regulating E-cadherin localization and clustering. Finally, actin also appeared more dispersed over the cytoplasm, with some clusters localized at the membrane, when compared to control cells, and the fluorescence signal is diminished at cell-cell contacts in the presence of estrogen.

## 4. Discussion

In this work we studied the influence of estrogen on the mechanical properties of MCF-7 cells, as a representative model of ER-positive breast cancer. Understanding the effects of estrogen on cellular mechanics can contribute to further elucidating the complex networks enabling tumor formation, cancer progression and metastasis. In this study we demonstrate that estrogen decreases the Young’s modulus and the viscoelastic mechanical properties of the cells. Further, we show altered cell-to-cell contacts in response to estrogen, in both AFM and confocal images, as well as E-cadherin localization, suggesting reduced cell-cell interactions.

Studies have shown that estrogen leads to a downregulation of E-cadherin at mRNA and protein levels in MCF-7 cells [49]. E-cadherin downregulation is thought to be one of the main reasons for loss of cell-cell-contacts and therefore tissue integrity in cancer [50]. However, others have reported that modulation of cell-cell adhesion by estrogen signaling occurs through E-cadherin membrane redistribution and adherens junction organization, without altering E-cadherin expression levels [7]. This conforms with our data, which show that treatment with estrogen leads to a less well-defined E-cadherin distribution at cell-cell contacts and to a redistribution of E-cadherin to the cytoplasm. In addition, both AFM and fluorescence images support the notion that adhesion between neighboring cells is weakened and cell borders appear less smooth when cells are in the presence of estrogen.

The cell softening observed by AFM has previously been associated with the capacity for invasion, as an adaptive mechanism to achieve increased motility [51,52]. On the other hand, use of anti-estrogens has been shown to induce an increase in cell stiffness, which coincided with the inhibition of breast cancer cell motility and the estrogen-dependent spatial reorganization of adherens junctions [7]. However, higher elasticity, which is associated with a stronger cytoskeleton, has also been associated with cancer cell invasion, suggesting that cell contractility and stiffness are not necessarily synonymous [53]. Indeed, a consistent increase of cancer cell viscosity has been observed during invasion into three-dimensional matrices [54]. Breast cancer progression and aggression has been associated with stroma stiffening and it has been shown that the more aggressive basal-like and HER2 tumor subtypes are stiffer than ER-positive breast tumors, which have a better prognosis [55]. Nevertheless, ER-positive tumors frequently develop resistance to hormone therapy [56], leading to increased aggressiveness and cancer stem cell content [57]. In fact, increased extracellular matrix stiffness has also been associated with increased resistance to chemotherapeutic drugs of breast cancer cells and with elevated expression of Sox2 and other stem cell factors in hepatocellular carcinoma [58,59]. Therefore, alterations in the mechanical properties of breast cancer when cells are exposed to different hormones during tumorigenesis, in hormone treatment and in acquisition of resistance to hormone therapy are expected and, thus, further investigation in this area is warranted.

## Figures and Tables

**Figure 1 materials-14-02897-f001:**
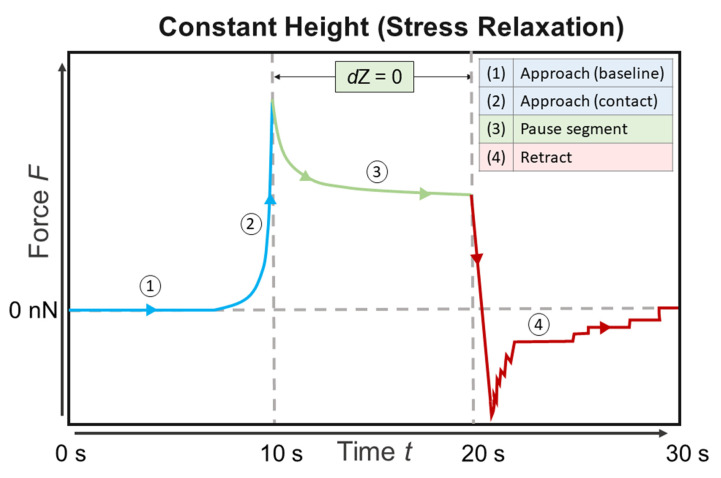
Schematic drawing of a force-time curve showing a stress relaxation experiment as performed in the atomic force microscopy (AFM) measurements.

**Figure 2 materials-14-02897-f002:**
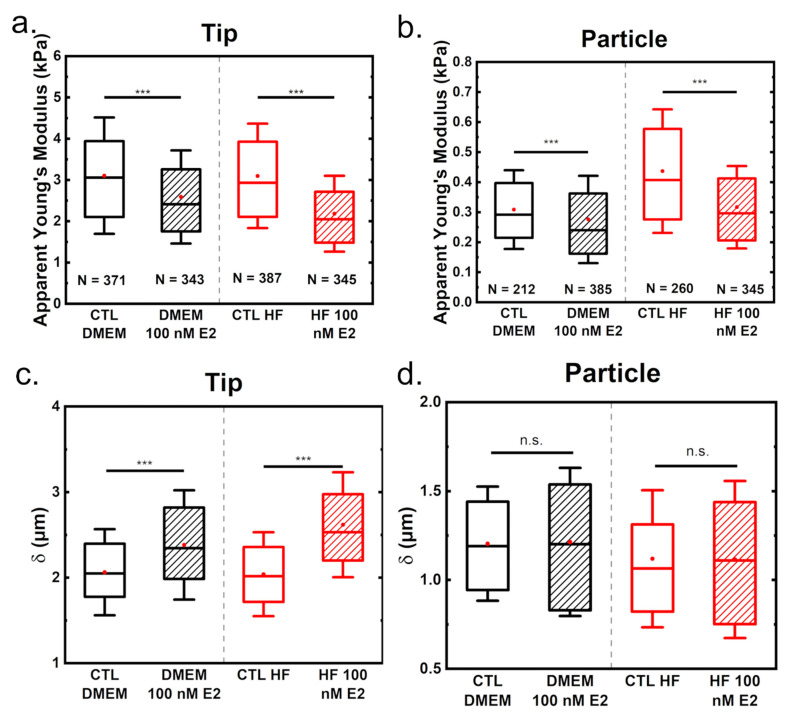
Influence of estrogen on the mechanical properties (apparent Young’s modulus for tips in (**a**) and particles in (**b**), and indentations at a force of 2 nN in (**c**) for tips and (**d**) for particles) for measurements performed with either tips or particles. An indentation depth of 500 nm was evaluated for Young’s modulus determination. N indicates the number of measurements considered. The boxplots indicate the data from the 25th to the 75th quantile, with the whisker being 1× the standard deviation, the red dot the mean value and the line in the middle the median. Statistical significances are reported as *** for *p* < 0.001.

**Figure 3 materials-14-02897-f003:**
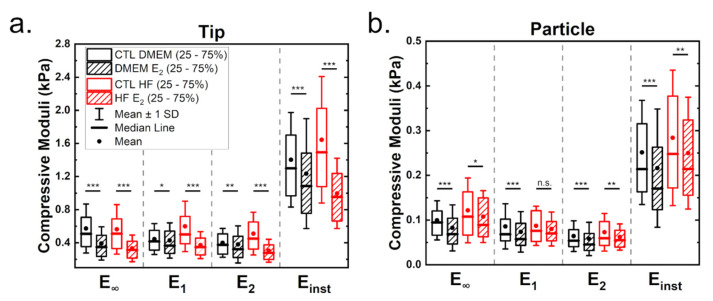
Viscoelastic mechanical properties determined from stress relaxation curves for measurements employing tips (**a**) and particles (**b**). The stress relaxation segment was recorded at a load of 2 nN (corresponding to an indentation of approximately either 2 or 1.25 µm, respectively) and a duration of 10 s. Statistical significances are reported as * for *p* < 0.05, ** < 0.01 and *** < 0.001.

**Figure 4 materials-14-02897-f004:**
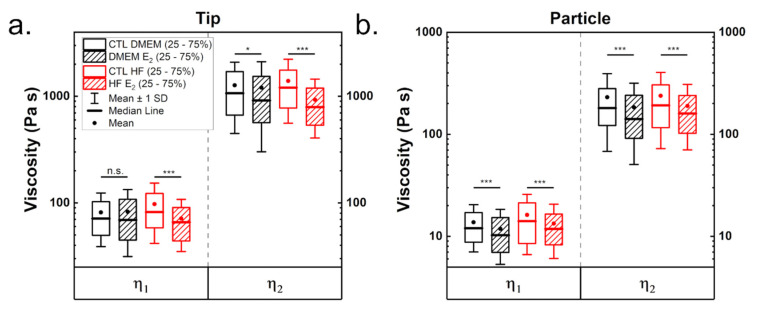
Viscosities determined from stress relaxation measurements when using a tip (**a**) or a particle (**b**) as an indenter. Equation (15) was used to calculate the shown values. Significances are reported as * for *p* < 0.05, and *** < 0.001.

**Figure 5 materials-14-02897-f005:**
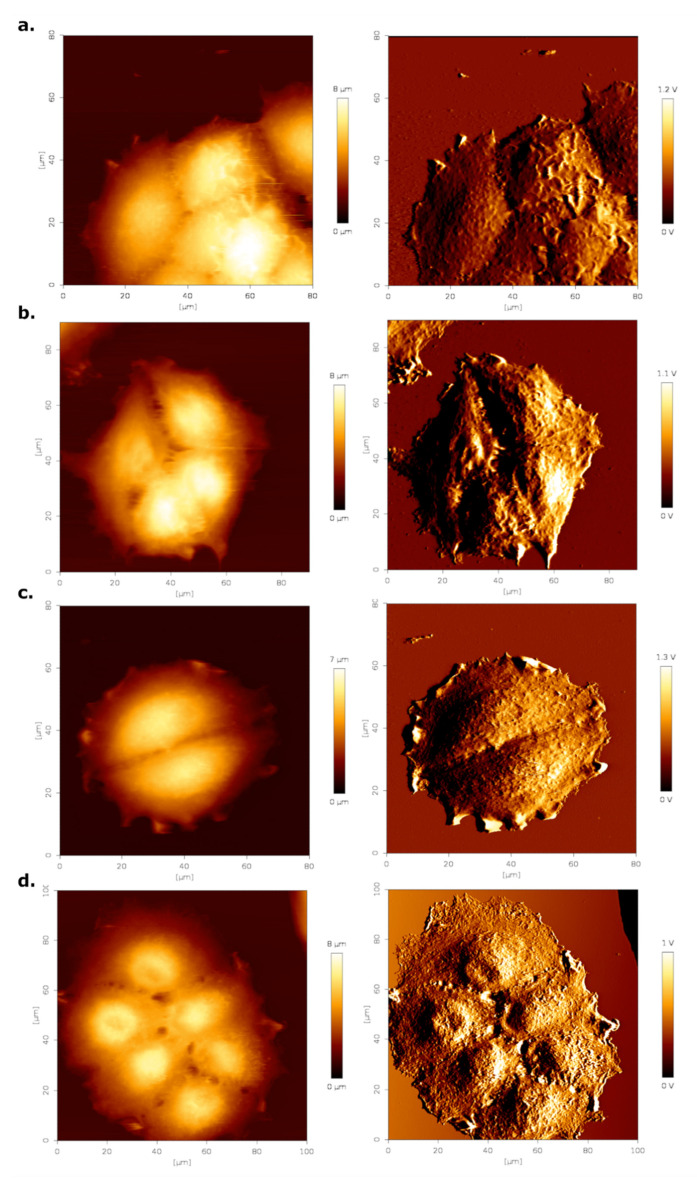
AFM contact mode images (height—left column, error—right column) of MCF-7 cells. (**a**) DMEM control, (**b**) DMEM treated with estrogen, (**c**) hormone-free medium control and hormone-free medium with estrogen (**d**). Measurements were performed in PBS at RT. Brighter pixel values correspond to a higher height/error value.

**Figure 6 materials-14-02897-f006:**
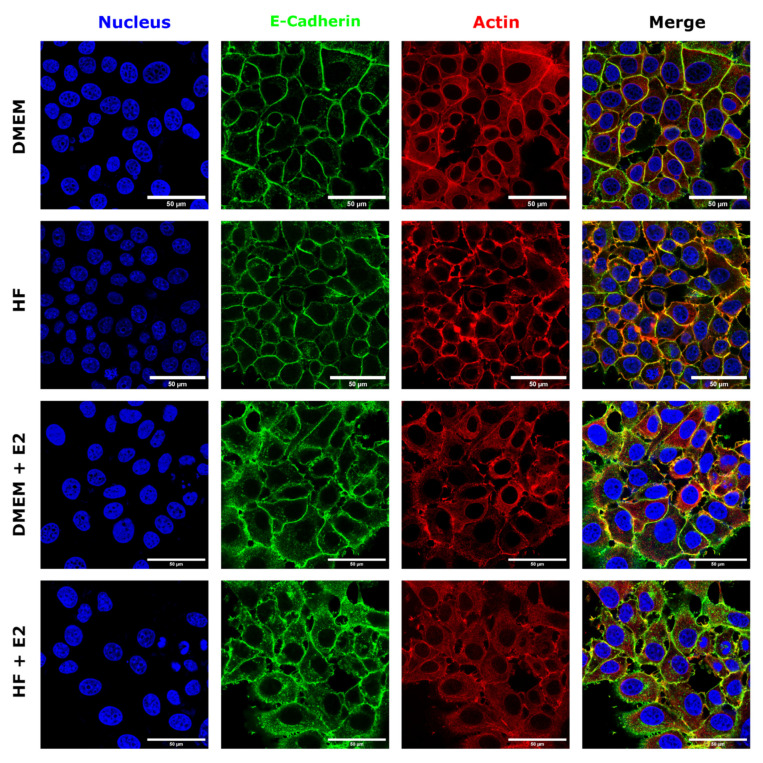
Confocal fluorescence microscopy images of MCF-7 cells grown either in DMEM, HF, DMEM supplemented with 100 nM estrogen or hormone-free medium enhanced with 100 nM estrogen. Nuclei are shown in blue, E-cadherin in green, actin in red. The scale bar corresponds to 50 µm.

**Table 1 materials-14-02897-t001:** Median values of the elastic and viscoelastic properties derived for measurements on MCF-7 cells with either tips or particles (CTL and estrogen-treated).

	Tip				Particle			
	CTL DMEM	DMEM E_2_	CTL HF	HF E_2_	CTL DMEM	DMEM E_2_	CTL HF	HF E_2_
*E_elastic_* (Pa)	3058.9	2410.5	2933.1	2046.8	308.7	275.9	436.7	316.6
*E_∞_* (Pa)	507.9	347.9	508.8	312.7	94.5	68.4	107.5	89.3
*E*_1_ (Pa)	416.2	364.7	501.0	347.8	68.0	57.3	75.9	69.9
*E*_2_ (Pa)	374.9	321.6	449.3	277.2	53.7	45.3	59.6	55.0
*E_inst_* (Pa)	1299.1	1085.3	1492.3	952.3	214.0	170.2	247.5	214.1
*η*_1_ (Pa s)	71.2	69.05	81.8	65.8	12.0	10.2	14.1	11.8
*η*_2_ (Pa s)	1067.7	912.5	1205.9	791.3	180.7	141.2	191.6	159.9

**Table 2 materials-14-02897-t002:** Mean values ± SE of cell height, average roughness and root mean square roughness determined from AFM contact mode height images. At least 25 cells were evaluated per treatment.

	Cell Height (µm)	*R_a_* (nm)	*R_q_* (nm)
CTL DMEM	5.25 ± 0.16	128 ± 3	166 ± 3
DMEM E_2_	5.98 ± 0.23	182 ± 6	220 ± 8
CTL HF	5.62 ± 0.19	139 ± 5	178 ± 5
HF E_2_	5.96 ± 0.20	147 ± 5	180 ± 6

## Data Availability

Data available on request from the corresponding author.

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
