# Peer review of "Estrogen Modulates Epithelial Breast Cancer Cell Mechanics and Cell-to-Cell Contacts"

_materials, 2021, doi:10.3390/ma14112897_

Round 1

Reviewer 1 Report

Estrogens are a group of steroid hormones playing the important role in regulation of a number of physiological processes. They contribute to cognitive health, bone health, the function of the cardiovascular system among others. However excessive exposure to estrogen is associated with increased risk of breast cancer.

The presented data suggest that estrogen alters cell-to-cell contacts in estrogen receptor-positive breast cancer cells and led to significant softening of the MCF-7 cells. This study

The manuscript is prepared very carefully with original data presented in details and documented with 6 regular figures and supplementary material. The references comprise of the newest papers as well as important older items. The detailed discussion emphasizes AFM techniques enables the examination and determination of adhesion between neighbouring cells and the characterisation of cell borders. Both are weakened in the presence of estrogen. This study describes some new roles of estrogen signalling in breast cancer and opens new area which need further investigation.

Minor comment:

  1. Figure 2: the legend is hardly readable.
  2. The confocal fluorescence microscopy: I am curious if you ever prepared images of the MCF-7 cells exposed to estrogen. Does the actin is directly diffused in the cytoplasm or presumably co-localise with intracellular organelles?

Author Response

Minor comment:

  1. Figure 2: the legend is hardly readable.
  2. The confocal fluorescence microscopy: I am curious if you ever prepared images of the MCF-7 cells exposed to estrogen. Does the actin is directly diffused in the cytoplasm or presumably co-localise with intracellular organelles?

We have changed Figure 2 so that now it is better readable. We have put the information that was in the legend (being the factors described that are shown in the boxplot) in the figure caption. Regarding the second questions: We have performed CLSM of the cells exposed to estrogen and followed the actin distribution (see Figure 6). We don’t see strong, pronounced changes in the actin distribution.

Reviewer 2 Report

  1. The authors are recommended to concise the introduction section instead of too general.
  2. The lines 254 to 259 are confusing, especially line 258. Please proofread all well.
  3. Please use legible fonts and axis values in figure 5. It is hard to read anything.
  4. Figure 6, all images need to be verified and rechecked for consistency.
  5. The authors are strongly recommended to highlight the conclusion of this study evidently.
  6. English must be improved throughout the manuscript. There are many awkward or grammatically incorrect expressions.

Author Response

  1. The authors are recommended to concise the introduction section instead of too general.

We have shortened the introduction and made it more concise.

  1. The lines 254 to 259 are confusing, especially line 258. Please proofread all well.

We have changed the passage aiming for it to be better understandable.

  1. Please use legible fonts and axis values in figure 5. It is hard to read anything.

We have re-arranged figure 5 to better show the axis values.

  1. Figure 6, all images need to be verified and rechecked for consistency.

We have re-checked the measurements done for Figure 6 (and also the control measurements not shown) for consistency.

  1. The authors are strongly recommended to highlight the conclusion of this study evidently.

We have tried to better highlight the conclusions of the study in the discussion section.

  1. English must be improved throughout the manuscript. There are many awkward or grammatically incorrect expressions.

All authors went through the manuscript again and improved the English.